# HER2 Upregulates ATF4 to Promote Cell Migration via Activation of ZEB1 and Downregulation of E-Cadherin

**DOI:** 10.3390/ijms20092223

**Published:** 2019-05-06

**Authors:** Peng Zeng, Shengnan Sun, Rui Li, Zhi-Xiong Xiao, Hu Chen

**Affiliations:** Center of Growth, Metabolism and Aging, Key Laboratory of Bio-Resource and Eco-Environment, Ministry of Education, College of Life Sciences, Sichuan University, Chengdu 610064, China; pengzengscu@163.com (P.Z.); shengnansun0626@163.com (S.S.); Lr804780807@163.com (R.L.); jimzx@scu.edu.cn (Z.-X.X.)

**Keywords:** HER2, ATF4, ZEB1, E-cadherin, cell migration

## Abstract

HER2 (human epidermal growth factor receptor 2) activation is critical in breast cancer development. HER2 promotes cell proliferation, angiogenesis, survival, and metastasis by activation of PI3K/Akt, Ras/MEK/ERK, and JAK/STAT pathways. However, beyond these signaling molecules, the key proteins underlining HER2-mediated metastasis remain elusive. ATF4 (Activating transcription factor 4), a critical regulator in unfolded protein response (UPR), is implicated in cell migration and tumor metastasis. In this study, we demonstrate that HER2 upregulated ATF4 expression at both mRNA and protein levels, resulting in cell migration increased. In addition, ATF4 upregulated ZEB1 (Zinc finger E-box-binding homeobox 1) and suppressed E-cadherin expression resulting in promoting cell migration. Restoration of E-cadherin expression effectively inhibited HER2- or ATF4-mediated cell migration. In addition, upregulated expression of ATF4 was found in HER2-positive breast cancer specimens. Together, this study demonstrates that ATF4-ZEB1 is important for HER2-mediated cell migration and suggests that ATF4-ZEB1 may be potential therapeutic targets for breast cancer metastasis.

## 1. Introduction

HER2/Neu (ErbB2) is a member of the ErbB family of receptor tyrosine kinases. *HER2* gene is frequently amplified in various human cancers. Overexpression of HER2 is found in approximately 20–30% of breast cancers, 28% of pulmonary adenocarcinoma, 17% of colorectal adenocarcinomas, and 11% of gastric adenocarcinomas [1]. HER2 overexpression is also found in ovarian and aggressive forms of uterine cancer [2,3]. HER2 forms homodimers or heterodimers with HER3/HER4 to activate its receptor tyrosine kinase activity and thereby triggering its downstream cascades involving in PI3K/Akt, Ras/MEK/ERK, and JAK/STAT in the regulation of a variety of cellular functions. Elevated HER2 expression is associated with poor prognosis of breast cancer patients including higher mortality, relapse, and metastasis [4]. HER2 has been well documented as an important therapeutic target in breast cancer treatment. Currently, three specific HER2-targeting antibodies, trastuzumab (Herceptin^TM^), pertuzumab (Perjeta^TM^), and T-DM1 (Kadcyla^TM^), are widely used to treat HER2-positive breast cancer patients [5].

ATF4, a member of ATF/CREB (activating transcription factor/cyclic AMP response element binding protein) family, belongs to the basic region-leucine zipper (bZip) transcription factors. ATF family members play critical roles in tumorigenesis and progression. Overexpression of ATF1 or ATF2 accelerates melanoma growth and metastasis [6,7]. ATF3 has a complex role in cancer development, while it binds to and stabilizes p53 thereby promoting apoptosis in untransformed mammary epithelial cells, it promotes breast cancer cell survival and enhances breast cancer cell mobility [8]. Furthermore, ATF3 appears to enhance TGFβ signaling-mediated EMT [9]. ATF5 enhances malignant glioma survival by stimulating transcription of MCL1, an antiapoptotic B cell leukemia-2 family member [10]. Similarly, ATF6 also promotes colorectal tumorigenesis and survival of dormant tumor cells in vivo [11,12]. Conversely, ATF7 suppresses tumorigenesis in mouse lymphoma models [13]. It has been reported that ATF4 is important in the development of adipogenesis, lipogenesis [14], osteogenesis, and in fatty acid and amino acid metabolism [15]. In addition, ATF4 plays a critical role in the cellular response to ER (endoplasmic reticulum) stress and oxidative stress. ER stress triggers unfolded protein response (UPR) through three parallel signaling branches: PERK/eIF2α, IRE1α/XBP1, and ATF6α [16]. Among these UPR signaling pathways, only PERK/eIF2α signaling upregulates ATF4 expression. Upon ER stress, PERK is activated through homodimerization and autophosphorylation, which in turn phosphorylates and inhibits eIF2α, a translation initiation factor, leading to a reduction of global protein synthesis but specifically accelerating the translation of ATF4 mRNA [17]. Expression of ATF4 is upregulated in cancers and has been shown to promote cell proliferation, survival, drug resistance, migration, and metastasis [18]. Recently, it has been shown that ATF4 stimulates expression of LAMP3 in facilitating breast cancer cell migration [19] and directly transactivates HO-1 (heme oxygenase 1) gene expression to inhibit anoikis and to promote tumor metastasis [20].

E-cadherin is a well-documented epithelial marker and plays an essential role in cell adhesion. Loss of E-cadherin is a key initial step in the process of epithelial-to-mesenchymal transition (EMT). E-cadherin expression is tightly regulated at genetic, epigenetic, transcriptional and post-translational levels during tumorigenesis [21]. At the transcriptional level, E-cadherin is repressed by several key EMT-promoting transcription factors (EMT-TFs), including ZEB1, ZEB2, Snail, Slug, Twist1/2, and E12/47, which directly bind to the E-boxes of E-cadherin gene promoter and repress its transcription [22]. Notably, ZEB1 interacts with its corepressor CtBP or BRG1 to repress E-cadherin expression, which in turn promotes tumor cell invasion and metastasis [23,24]. Interestingly, it is reported that Slug is a direct transcriptional activator at E-boxes of the ZEB1 promoter [25] and that Snail1 and Twist cooperate in the induction of ZEB1 expression [26]. In addition, there is a reciprocal negative feedback loop between miR-200 family members (miR-200a, miR-200b, miR-200c, miR-141, and miR-429) and ZEB1. Increasing the expression of ZEB1 results in repression of miR-200 members, resulting in decreased RNAi on the ZEB1 mRNA. Conversely, increasing the expression of miR-200 family members decreases ZEB1 levels, which in turn increases miR-200 expression [27]. Notably, TGFβ, NF-κB, and hypoxia have been implicated in upregulating ZEB1 expression [28,29].

In this study, we demonstrated that HER2 upregulates ATF4 expression to promote cell migration. Knockdown of ATF4 completely reverses HER2-induced cell migration. ATF4 upregulates ZEB1, resulting in increased cell migration. Restoration of E-cadherin effectively inhibits ATF4-induced cell migration. Together, our findings indicate that ATF4-ZEB1-E-cadherin axis is critical for HER2-mediated cell migration and that targeting of ATF4-ZEB1 may be a potential therapeutic strategy for HER2-mediated tumor metastasis.

## 2. Results

### 2.1. HER2 Induces Cell Migration through Upregulation of ATF4 and Downregulation of E-Cadherin

HER2 activation is critical in breast cancer development and ATF4 has been shown to play a role in breast cancer migration. To investigate whether there is an intrinsic relationship between HER2 and ATF4, we expressed HER2 in human non-transformed mammary epithelial MCF10A cells. As shown in Figure 1A,B, expression of HER2 led to spindle-like cell morphology and scattered cell growth as well as significantly increased cell migration. In addition, HER2 expression led to a markedly decrease of E-cadherin expression, concomitant with an upregulation of ATF4 protein levels (Figure 1C). To investigate the role of ATF4 in HER2-induced cell migration, we knocked down ATF4 by shRNA in MCF10A cells with HER2 overexpressed. As shown in Figure 1D, knockdown of ATF4 led to restoring E-cadherin expression which was suppressed by HER2. Notably, knockdown of ATF4 completely inhibited HER2-induced scattered cell growth (Figure 1E) and effectively suppressed HER2-induced cell migration (Figure 1F). To address whether cell proliferation contributes to the increased migration, we detected cell proliferation by using Real-Time Cell Analyzer. As shown in Figure 1G, expression of HER2 significantly enhanced MCF10A cell proliferation only after 32 h seeding, suggesting that cell proliferation is unlikely to contribute to the increased cell migration at which we performed transwell assays. Next, we investigated the role of E-cadherin downregulation in HER2-induced cell migration. As shown in Figure 1H–J, restoration of E-cadherin expression completely reversed HER2-induced scattered cell growth and also significantly inhibited HER2-mediated cell migration. Together, these data indicate that ATF4 and E-cadherin play a critical role in HER2-induced cell migration.

### 2.2. HER2 Upregulates ATF4 Transcription and Promotes its Protein Expression in an mTOR-Dependent Manner

To investigate the mechanism in which HER2 upregulates ATF4, we measured steady-state levels of ATF4 mRNA by Q-PCR and protein half-life by cycloheximide (CHX) treatment. As shown in Figure 2A–C, HER2 significantly upregulated steady-state ATF4 mRNA levels, but there was little effect on ATF4 protein half-life. In addition, inhibition of mTOR by rapamycin, as evidenced by the reduction of phosphorylated S6K, completely blocked HER2-induced upregulation of ATF4 protein expression (Figure 2D), suggesting that HER2 promotes ATF4 protein expression via mTOR signaling.

### 2.3. ATF4 Promotes Cell Migration via Suppressing E-Cadherin Expression

Our aforementioned data indicate that knockdown of ATF4 inhibited HER2-induced cell migration, concomitant with upregulated E-cadherin protein expression. We then investigated whether there is a causative relationship between ATF4 and E-cadherin. As shown in Figure 3A–C, ectopic expression of ATF4 in MCF10A cells led to a marked decrease of E-cadherin protein (Figure 3A) and accelerated cell migration (Figure 3C). Expression of ATF4 also downregulated E-cadherin expression in breast cancer MCF7 cells (Figure 3B). On the contrary, the knockdown of ATF4 significantly upregulated E-cadherin protein expression (Figure 3D), resulting in cell migration significantly inhibited (Figure 3E). Furthermore, restoration of E-cadherin expression effectively reversed ATF4-induced cell migration (Figure 3F,G).

We then investigated the molecular mechanism in which ATF4 downregulates E-cadherin expression. As shown in Figure 3H, expression of ATF4 significantly downregulated steady-state E-cadherin mRNA levels. Notably, ATF4 markedly increased ZEB1 expression at both mRNA and protein levels (Figure 3I,J), and decreased E-cadherin expression, in keeping with the well-documented role of ZEB1 in the suppression of E-cadherin [24]. In addition, knockdown of ZEB1 significantly increased E-cadherin expression which was suppressed by ATF4 (Figure 3K). Taken together, these results indicate that ATF4 promotes cell migration via upregulation of ZEB1, which in turn inhibits E-cadherin expression.

### 2.4. HER2, ZEB1, and ATF4 Expression is Correlated in Human Breast Cancers

To examine the clinical relevance of HER2, ATF4, and ZEB1, we analyzed the expression of these genes in invasive breast carcinoma using TCGA databases. As shown in Figure 4A, ATF4 expression was significantly upregulated in HER2-positive breast cancers. In addition, ZEB1 expression also substantially increased in HER2-positive breast specimens (Figure 4B). Together, these clinical data indicate that ATF4 plays an important role in HER2-positive breast cancer patients.

## 3. Discussion

It has been well documented that activation of HER2, as exemplified by gene amplification, is a major driving force for the development of breast cancer, lung cancer, colorectal cancer, gastric cancer, ovarian cancer and uterine cancer. HER2 promotes cell proliferation, survival, migration, invasion, and angiogenesis by the activation of PI3K/Akt, Ras/MEK/ERK, and JAK/STAT pathways. We have previously demonstrated that HER2 targets and suppresses ΔNp63α gene transcription to promote tumor metastasis [30]. In this study, we found that ATF4, a critical regulator in ER stress response, is a novel target of HER2, as evidenced by: (1) expression of HER2 upregulates ATF4 expression at both mRNA and protein levels; (2) knockdown of ATF4 inhibits HER2-induced cell migration; and (3) expression of ATF4 is correlated with HER2-positive breast cancer specimens. Notably, *PI3KCA* hotspot mutation (H1047R and E545K) activated PDK1-RSK2 pathway, resulting in USP8 binding to ATF4 and stabilizing its protein to reprogram glutamine metabolism [31]. In addition, activation of PI3K/AKT by K-Ras stimulates expression of NRF2, which in turn transactivates ATF4 in promoting cancer cell survival and growth upon amino acid deprivation [32]. It is, therefore, worth to investigate whether HER2 activates ATF4 through RSK2 and NRF2. Recently, it has been reported that PI3K activates mTORC1 and mTORC2 to upregulate ATF4 [33,34]. Consistently, our data showed that blocking mTORC1 by rapamycin inhibits HER2-induced ATF4 upregulation, indicating that HER2 likely activates PI3K-mTORC1 to facilitate ATF4 protein expression.

In this study, we found that downregulation of E-cadherin is pivotal in ATF4-induced cell migration, in keeping with numerous reports that loss of E-cadherin expression is critical for cell migration and tumor metastasis [35]. Our data clearly demonstrated that ATF4 promotes cell migration through repressing E-cadherin expression at both mRNA and protein levels. Restoration of E-cadherin completely reversed ATF4-mediated cell migration. Thus, E-cadherin is a novel target of ATF4 responsible for HER2-induced cell migration. At the molecular level, we showed that ATF4 significantly upregulates the mRNA and protein levels of ZEB1, which is a critical transcriptional repressor of E-cadherin and promotes EMT and cancer metastasis [24]. Therefore, our study reveals a novel regulatory mechanism with which HER2-ATF4-ZEB1-E-cadherin signaling axis regulates cancer cell migration and tumor metastasis.

Over the course of tumorigenesis, tumor cells can only proliferate and metastasize if they can cope with the challenge of limited sources of oxygen, glucose, and amino acids. These limitations often lead to ER stress to activate unfolded protein response (UPR). ATF4 is a critical transcriptional factor of UPR [36] and is frequently upregulated in various human tumors and cancers. Oncogenic signals, including Myc, *PIK3CA* hotspot mutations (H1047R and E545K) and *KRAS* hotspot mutation (G12V), upregulate ATF4 to promote cancer survival. In this study, we showed that upregulated ATF4 expression is correlated with HER2-positive breast cancer specimens, suggesting that the ATF4 may play a role in HER2-mediated breast cancer development. It is conceivable that ATF4-ZEB1 could be a potential therapy target for breast cancer treatment.

## 4. Materials and Methods

### 4.1. Cell Culture, Cell Morphology, and Cell Proliferation Assays

MCF10A cells were maintained in 1:1 mixture of DMEM and F12 medium (Invitrogen, Carlsbad, CA, USA), 5% horse serum, 100 ng/mL cholera toxin (Sigma, St Louis, MO, USA), 10 μg/mL insulin (Sigma, St Louis, MO, USA), 20 ng/mL epidermal growth factor (Invitrogen, Carlsbad, CA, USA), and 500 ng/mL (95%) hydrocortisone (Sigma). MCF7 cells were maintained in MEM supplemented with 10% fetal bovine serum (Hyclone, Logan, UT, USA) and 5 μg/mL insulin (Sigma, St Louis, MO, USA). All cells were grown in media containing 100 U/mL penicillin and 100 μg/mL streptomycin (GIBCO, Rockville, MD, USA) at 37 °C in a humidified incubator under 5% CO_2_. To assess cell morphology, cells were seeded at low confluence and grew for 7–10 day, then fixed with 4% paraformaldehyde and stained with 0.1% Crystal violet in 70% ethanol, and photographed by light microscope. To test cell proliferation, MCF10A cells were seeded at a density of 5000 cells/well in Real-Time Cell Analyzer (RTCA, ACEA Biosciences, Inc., San Diego, CA, USA), the cell index signals were recorded every 1 h for 48 h.

### 4.2. Plasmid Construction and Lentivirus Infection

Human ATF4 and E-cadherin (encoded by *CDH1* gene) were obtained by PCR with specific primers (ATF4-F:5′-ATGACCGAAATGAGCTTCCTG-3′, ATF4-R: 5′-CCCTAGGGGACCCTTTTCTTC-3′; CDH1-F: 5′-TGCTCTAGAGCCACCATGGGCCCTTGGAGCCGC-3′, CDH1-R: 5′-CCGCTCGAGCTAGTCGTCCTCGCCGCCTC-3′), then cloned into the pLVX-puro vector. Human HER2 was subcloned into Plenti-M3 vector from pEnter-HER2 construct (CH805581, Vigene Biosciences, Jinan, China) with the forward primer: 5′-TGCTCTAGAATGGAGCTGGCGGCCTTGTG-3′ and reverse primer: 5′-CCGCTCGAGTTACACTGGCACGTCCAGAC-3′. shRNA lentiviral constructs were cloned in the pLKO.1 vector. shRNA targeting sequences used are: shC, 5′-GAAGCAGCACGACTTCTTC-3′; shATF4-#1, 5′-GTCCTCCACTCCAGATCATTC-3′; shATF4-#2, 5′-TGGATGCCCTGTTGGGTATAG-3′; shZEB1-#1, 5′-CCTACCACTGGATGTAGTAAA-3′; shZEB1-#2, 5′-CCTCTCTGAAAGAACACATTA-3′. All plasmids were confirmed by DNA sequencing. Lentivirus were generated by transfectingHEK-293T cells at 80–90% confluence with pMD2.G and psPAX2 packaging plasmids using Lipofectamine 2000 (Invitrogen). Virus was harvested after 48 h post-transfection, filtered, and used for cell infection at 30% confluence with 10 µg/mL polybrene.

### 4.3. Western Blot Analysis

Cells were lysed in EBC250 lysis buffer (250 mM NaCl, 25 mM Tris, 50 mM NaF, 0.5% Nonidet P-40, and supplemented with 0.5 mM Na_3_VO_4_, 0.2 mM PMSF, 20 μg/mL aprotinin, and 10 μg/mL leupeptin, pH 8.0). Equal amounts of total proteins were separated by SDS-PAGE and transferred to PVDF membrane (Millipore, Darmstadt, Germany). The membrane was blocked in 4% milk in TBST for 1 h, hybridized to an appropriate primary antibody at 4 °C for overnight and HRP-conjugated goat anti-rabbit IgG (1:3000, sc-2004, Santa Cruz Biotech, Shanghai China) for 1 h at room temperature, then detected by chemiluminescence (WBKLS0500, Millipore, Darmstadt, Germany). Specific antibodies against ATF4 (1:1000, Cat. no. 11815), HER2 (1:1000, Cat. no. 2242), Vimentin (1:1000, Cat. no. 5741), GAPDH (1:3000, Cat. no. 5174), p-S6K (1:1000, Cat. no. 2211), S6K (1:1000, Cat. no. 2317), ZEB1 (1:1000, Cat. no. 3396) were purchased from Cell Signaling Technology (Danvers, MA, USA). E-cadherin (1:1000, ab40772) and N-cadherin (1:1000, ab2247-1) were purchased from Abcam (Cambridge, MA, USA).

### 4.4. Transwell Assays

Transwell assays for cell migration were performed in transwell inserts with a 6.5mm, 8.0 μm-pore polycarbonate membrane (BD Biosciences, San Jose, CA, USA). Cells were suspended in serum-free medium and seeded into the upper chamber (MCF10A, 5 × 10^4^ cells per chamber). The lower chamber contained 600 μL complete medium. Cells were incubated for 24 h at 37 °C in a humidified incubator under 5% CO_2_, then removed the non-migration cells on the inside of the chamber carefully with cotton swabs. The chambers were fixed with 4% paraformaldehyde and stained with 0.1% Crystal violet in 70% ethanol for 15 min, and photographed by light microscope and analyzed by Image J.

### 4.5. Q-PCR

Total RNA was extracted from cells by RNeasy Plus Mini Kit (cat# 74134, QIAGEN, Hilden, Germany) according to the manufacturer’s protocol. RNA was reverse-transcribed into cDNAs. Q-PCR was performed for E-cadherin (F: 5′-GGATGTGCTGGATGTGAATG-3′, R: 5′-CACATCAGACAGGATCAGCAGAA-3′); ATF4 (F: 5′-GCAAAAGTAAAGGGTGAGAAACTGG-3′, R: 5′-CTCGTTCTTCTTTTCCAGCTCTTTG-3’); ZEB1 (F: 5′-ACCCTTGAAAGTGATCCAGC-3′, R: 5′-CATTCCATTTTCTGTCTTCCGC-3’), and GAPDH (F: 5′-GGGGAGCCAAAAAGGGTCATCATCT-3′, R: 5′- GAGGGGCCATCCACAGTCTTCT-3′). Q-PCR assays performed in CFX96 Real-Time System (Bio-Rad, Hercules, CA, USA) using SYBR Green Supermix (Bio-Rad, Saint-Laurent, QC, Canada) at 95 °C for 3 min, followed by 40 cycles at 95 °C for 5 s and 60 °C for 15 s. Transcriptional levels of genes were normalized to GAPDH which were expressed stable and similar in all samples. The results were analyzed using 2^−ΔΔ^*^C^*^t^ method.

### 4.6. Protein Half-life

Cells were treated with 20 μg/mL cycloheximide (Sigma) and collected at the indicated time intervals. ATF4 protein levels were determined by Western blot analyses, quantified by densitometry from three independent experiments, and normalized to GAPDH.

### 4.7. Bioinformatics Analysis

TCGA database (cBioPortal, www.cbioportal.org) were used to analyze breast invasive carcinoma (TCGA Provisional, 1105 samples) for ATF4 and ZEB1 expression in HER2-positive and negative samples.

### 4.8. Statistical analysis

All statistical analyses were performed using SPSS 16 software (SPSS Inc., Chicago, IL, USA). The data were presented as means ± SD. Comparisons between two groups were performed using the two-tailed unpaired Student’s *t*-test. The differences between clinical data were compared after homogeneity tests. *p*-values less than 0.05 were considered statistically significant.

## Figures and Tables

**Figure 1 ijms-20-02223-f001:**
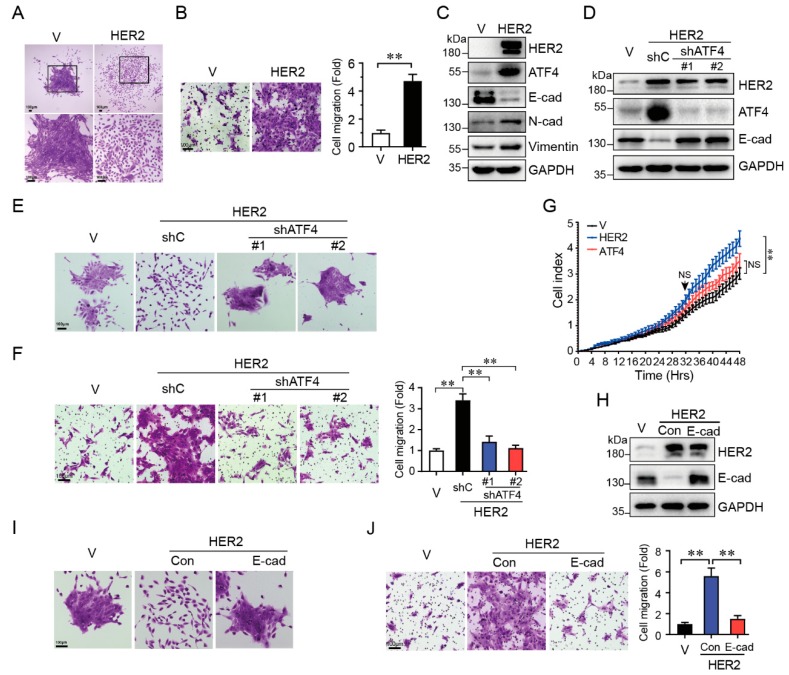
HER2 induces scattered cell growth and cell migration through upregulation of ATF4 and downregulation of E-cadherin expression. MCF10A cells stably expressing HER2 (**A**–**C**) were infected with a recombinant lentivirus expressing specific shRNA targeting to ATF4 (**D**–**F**) or a recombinant lentivirus expressing E-cadherin (**H–J**). Cells were then subjected to cell morphology observations (**A**,**E**,**I**), transwell assays (**B**,**F**,**J**), and Western blot analyses (**C**,**D**,**H**). MCF10A cells stably expressing HER2 or ATF4 were subjected to cell proliferation assays by Real-Time Cell Analyzer (**G**). Results are representative of three independent experiments. Data were presented as means ± SD; scale bars: 100 μm; NS: Not significant; **: *p* < 0.01.

**Figure 2 ijms-20-02223-f002:**
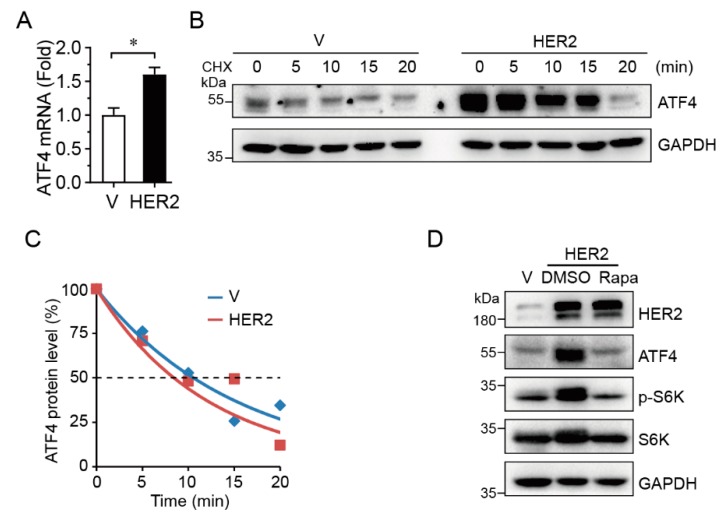
HER2 upregulates ATF4 expression at both transcriptional and translational levels. MCF10A-HER2 cells were subjected to Q-PCR analyses (**A**), protein half-life assay and the relative ATF4 protein expression levels were quantitated as described in the Materials and Methods (**B**–**C**), or were treated with or without rapamycin (100 nM) prior to Western blot analyses (**D**). The Q-PCR data were derived from three independent experiments; *: *p* < 0.05.

**Figure 3 ijms-20-02223-f003:**
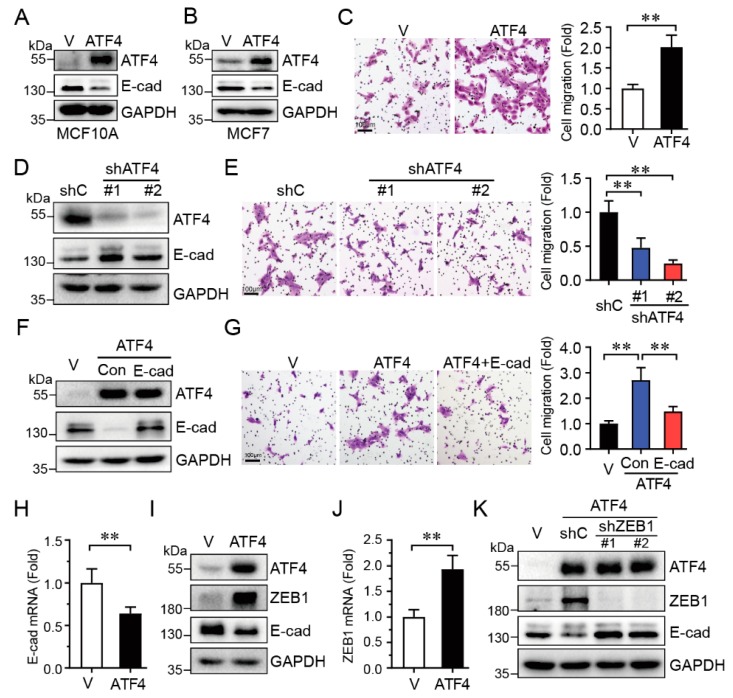
ATF4 promotes cell migration via suppression of E-cadherin expression. MCF10A-ATF4 cells were subjected to Western blot analyses (**A**,**I**), transwell assays (**C**), and Q-PCR assays (**H**,**J**). MCF7-ATF4 cells were subjected to Western blot analyses (**B**). MCF10A cells stably expressing specific shRNA targeting ATF4 or a control (shC), were subjected to Western blot analyses (**D**) or transwell assays (**E**). (**F**,**G**) MCF10A-ATF4 cells infected with a lentivirus expressing E-cadherin were subjected to Western blot analyses (F) or transwell assays (G). MCF10A-ATF4 cells infected with a lentivirus expressing shZEB1-#1, shZEB1-#2 or a control (shC), were subjected to Western blot analyses (**K**). Data were analyzed from three independent experiments and were presented as means ± SD; scale bars: 100 μm; **: *p* < 0.01.

**Figure 4 ijms-20-02223-f004:**
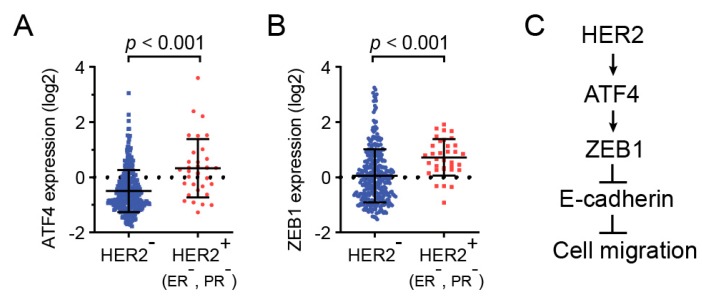
HER2, ZEB1, and ATF4 expression were analyzed in human breast cancers. mRNA levels of ATF4 or ZEB1 in HER2-positive or -negative breast cancer biopsy samples were analyzed using TCGA databases (**A**,**B**). A schematic model for the role of ATF4 in HER2-mediated cell migration (**C**).

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
