# Peer review of "HER2 Upregulates ATF4 to Promote Cell Migration via Activation of ZEB1 and Downregulation of E-Cadherin"

_ijms, 2019, doi:10.3390/ijms20092223_

Reviewer 1 Report

The manuscript entitled ‘HER2 upregulates ATF4 to promote cell migration via activation of ZEB1 and downregulation of E-cadherin’ by Zeng P and co-authors investigates the importance of ATF4 in breast cancer initiation and progression. The study includes both patient data and data derived from cell-based in vitro studies. The authors show convincing data on increased expression of ATF4 downstream of HER2, also resulting in increased expression of ZEB1 and suppression of E-cadherin, suggesting involvement of ATF4 in EMT induction. Expression of HER2, ATF4 and ZEB1 is also analyzed in samples from patients with advanced breast cancer using data from TCGA and Oncomine. The Materials and Methods section is clear and contains sufficient information regarding the methods included. Although the data presented are generally convincing, there are some points that should be considered regarding this manuscript:

Major points:

The non-transformed MCF10A breast epithelial cell line is used and analyzed throughout the manuscript, to investigate breast cancer initiation and progression. It’s then very surprising to me why the authors chose to compare the findings from MCF10A to a colorectal cancer cell line in Section 2.3. There are dozens of breast cancer cell lines commercially available that display different differentiation states and cancer stages to choose from for a more appropriate comparison, and there is no rationale behind why the HCT116 cell line was used instead.

Secondly, when analyzing HER2, ZEB1 and ATF4 expression in the breast cancer patient cohorts, the dataset from Oncomine is used for HER2 and ZEB1, whereas the dataset from TCGA is used for ATF4 analysis. There is no explanation for why ATF4 was analyzed using a different dataset.

In Fig. 1A-B, HER2 induction is said to induce a spindle-like cell morphology but the photos included are too small/zoomed out to clearly reveal these possible differences.

The statistical analyses are performed using Student’s t-test throughout the manuscript. This statistical method is very common, but demands that the variance is equal between the datasets that are being compared. Have homogeneity tests been performed on all datasets included?

For qPCR, GAPDH has been used as the only reference gene. It should be verified by the authors and stated in the manuscript that the Ct values for GAPDH were similar for all samples analyzed.

Additional points:

General: The manuscript should be read carefully, to correct for writing errors and misspellings.

The introduction about ATF and its actions is a bit scarce. The information given in the discussion using refs 22-27 could, at lest partly, be moved to the Introduction section.

It could be of interest for readers to get some more information regarding other ATFs and whether there is any redundancy between them. Are any other ATFs involved in breast cancer initiation/progression?

Author Response

Reviewer 1

Comments and Suggestions:

The manuscript entitled ‘HER2 upregulates ATF4 to promote cell migration via activation of ZEB1 and downregulation of E-cadherin’ by Zeng P and co-authors investigates the importance of ATF4 in breast cancer initiation and progression. The study includes both patient data and data derived from cell-based in vitro studies. The authors show convincing data on increased expression of ATF4 downstream of HER2, also resulting in increased expression of ZEB1 and suppression of E-cadherin, suggesting involvement of ATF4 in EMT induction. Expression of HER2, ATF4 and ZEB1 is also analyzed in samples from patients with advanced breast cancer using data from TCGA and Oncomine. The Materials and Methods section is clear and contains sufficient information regarding the methods included. Although the data presented are generally convincing, there are some points that should be considered regarding this manuscript:

Major points:

-The non-transformed MCF10A breast epithelial cell line is used and analyzed throughout the manuscript, to investigate breast cancer initiation and progression. It’s then very surprising to me why the authors chose to compare the findings from MCF10A to a colorectal cancer cell line in Section 2.3. There are dozens of breast cancer cell lines commercially available that display different differentiation states and cancer stages to choose from for a more appropriate comparison, and there is no rationale behind why the HCT116 cell line was used instead.

Response: We appreciated the reviewer's comments. Originally, we wanted to show that expression of ATF4 affecting E-cadherin expression is not cell type specific, it does happen in colon cancer HCT116 cells. However, as the reviewer's suggested, we performed similar experiment in breast cancer MCF7 cells and reached same conclusion. These data were incorporated in the Fig. 3B of the revised manuscript.

-Secondly, when analyzing HER2, ZEB1 and ATF4 expression in the breast cancer patient cohorts, the dataset from Oncomine is used for HER2 and ZEB1, whereas the dataset from TCGA is used for ATF4 analysis. There is no explanation for why ATF4 was analyzed using a different dataset.

Response: We appreciated the reviewer's constructive suggestion. Accordingly, we have re-analyzed ATF4 and ZEB1 expression in HER2-positive breast cancer specimen using the same TCGA dataset. The results are now presented as Figure 4A and 4B in the revised manuscript.

-In Fig. 1A-B, HER2 induction is said to induce a spindle-like cell morphology but the photos included are too small/zoomed out to clearly reveal these possible differences.

Response: We appreciated the reviewer's comments. In the revised Figure 1A, the larger visions of the microscopic fields reflecting cell morphology were used. The Figure 1B were the photos of migrated cells on the membrane using transwell analysis, they were not meant to exhibit cell morphology.

-The statistical analyses are performed using Student’s t-test throughout the manuscript. This statistical method is very common, but demands that the variance is equal between the datasets that are being compared. Have homogeneity tests been performed on all datasets included?

Response: We appreciated the reviewer's comments and suggestions. As suggested, we performed quantitative analyses of Transwell and QPCR data using the two-tailed unpaired Student’s t test. For clinical data, we compared the differences after homogeneity tests in SPSS 16 software and showed the original analysis results below. In the revised manuscript, we clarified in “Statistical analysis” of “Materials and Methods” section.

-For qPCR, GAPDH has been used as the only reference gene. It should be verified by the authors and stated in the manuscript that the Ct values for GAPDH were similar for all samples analyzed.

Response: We performed Q-PCR and compared the Ct in all samples. The Ct values of GAPDH were similar in all samples. We have clarified this issue in revised "Materials and Methods".

Additional points:

-General: The manuscript should be read carefully, to correct for writing errors and misspellings.

Response: We apologize for the spelling mistakes and writing errors. We have carefully proofread this revised manuscript to minimize those errors.

-The introduction about ATF and its actions is a bit scarce. The information given in the discussion using refs 22-27 could, at least partly, be moved to the Introduction section.

Response: We agree. In the revised manuscript, we have added substantially more information about ATF family and have introduced the role of ATF4 in cell migration and tumor metastasis in the section of "Introduction".

-It could be of interest for readers to get some more information regarding other ATFs and whether there is any redundancy between them. Are any other ATFs involved in breast cancer initiation/progression?

Response: In the revised manuscript, we have added substantially more information about ATF family with regard to their roles in breast cancer development.

Reviewer 2 Report

The authors show that ATF4 promotes cell migration through repressing E-cadherin expression and its restoration completely revert ATF-mediated cell migration.

All conclusions of this work are well supported by the experiments, but some points need to be addressed for publication.

Major points:

-The authors use only one shRNA for knocking down ATF4 and for E-cadherin expression, given that this is a critical point the authors should use in parallel also a different shRNA.

-The cell migration experiment takes 24 hrs; the authors should check that the cell migration effect is not influenced by a difference in cell proliferation rate.

-the authors have to show also the basal migration (w/o chemoattractant), indicating the raw data of migrating cells (before to normalization).

Minor points:

- It could be worth to analyse also the malignant brest carcinoma cells (i.e MCF-7 or MD-MB-231) and to compare the results to normal MCF-10A cells.

Author Response

Reviewer 2

Comments and Suggestions:

The authors show that ATF4 promotes cell migration through repressing E-cadherin expression and its restoration completely revert ATF-mediated cell migration.

All conclusions of this work are well supported by the experiments, but some points need to be addressed for publication.

Major points:

-The authors use only one shRNA for knocking down ATF4 and for E-cadherin expression, given that this is a critical point the authors should use in parallel also a different shRNA.

Response: We agree. Accordingly, we have performed a new set of experiments using a second and different shRNA against ATF4. Our data indicate that expression of shATF4-#2, like shATF4-#1, effectively rescued E-cadherin expression, scattered cell growth and cell migration, all of which was induced by overexpression of HER2 (revised Figure 1D-1F).

-The cell migration experiment takes 24 hrs; the authors should check that the cell migration effect is not influenced by a difference in cell proliferation rate.

Response: This is an excellent suggestion. We performed a new experiment to assess cell proliferation of MCF10A cells stably expressing ATF4 or HER2 using Real-Time Cell Analyzer (RTCA, ACEA Biosciences, Inc.). The results (see below) showed that the cell proliferation rates were not significantly altered (NS: not significant) during the course of 24 hours, which, importantly, were the time points for our transwell-based cell migration assays. We have made this point clear in the revised manuscript (Figure 1G).

-the authors have to show also the basal migration (w/o chemoattractant), indicating the raw data of migrating cells (before to normalization).

Response: We think that this is a valuable suggestion. Accordingly, we performed a new set of transwell assays. In the control experimental setting, the lower chamber contained just DMEM-F12 medium. By contrast, lower chamber contained DMEM-F12 medium (supplemented with 5% horse serum, insulin, cholera toxin, EGF, hydrocortisone to serve as chemoattractant). As shown below, MCF10A cells could not migrate through transwell membrane in the lower chamber with no chemoattractant. By sharp contrast, MCF10A cells vigorously migrated through transwell membrane with chemoattractant, which was greatly enhanced by overexpression of HER2.

Minor points:

- It could be worth to analyse also the malignant brest carcinoma cells (i.e MCF-7 or MD-MB-231) and to compare the results to normal MCF-10A cells.

Response: We performed a new experiment to examine the effect of ATF4 on E-cadherin expression in MCF7 cells (Revised Figure 3B).

Round  2

Reviewer 1 Report

The authors have responded to the comments within the previous review in a satisfactory way, and the manuscript has clearly been improved. Some writing errors and misspellings are still present, e.g. line 77, E-cadhetin; line 156, specimen -> specimens; line 159-161, the sentence doesn’t really make sense; line 177, though -> through; line 270, should be ‘HER2-positive and negative samples’. Besides the need for additional language corrections, the manuscript has, in my opinion, reached a scientific quality acceptable for publication in International Journal of Molecular Sciences.

Reviewer 2 Report

The authors included all suggestions and made suggested experiments. The maniscript is improved and in this form it is suitable for publication.